# Multiparametric Magnetic Resonance Imaging Findings of the Pancreas: A Comparison in Patients with Type 1 and 2 Diabetes

**DOI:** 10.3390/tomography11020016

**Published:** 2025-02-07

**Authors:** Mayumi Higashi, Masahiro Tanabe, Katsuya Tanabe, Shigeru Okuya, Koumei Takeda, Yuko Nagao, Katsuyoshi Ito

**Affiliations:** 1Department of Radiology, Yamaguchi University Graduate School of Medicine, Yamaguchi 755-8505, Japan; m-tanabe@yamaguchi-u.ac.jp (M.T.); itokatsu@yamaguchi-u.ac.jp (K.I.); 2Division of Endocrinology, Metabolism, Hematological Sciences and Therapeutics, Yamaguchi University Graduate School of Medicine, Yamaguchi 755-8505, Japan; ktanabe@yamaguchi-u.ac.jp (K.T.);; 3Health Administration Center, Organization for Education and Student Affairs, Yamaguchi 753-8511, Japan; okuya@yamaguchi-u.ac.jp

**Keywords:** pancreas, diabetes mellitus, magnetic resonance imaging

## Abstract

Background/Objectives: Diabetes-related pancreatic changes on MRI remain unclear. Thus, we evaluated the pancreatic changes on MRI in patients with both type 1 diabetes (T1D) and type 2 diabetes (T2D) using multiparametric MRI. Methods: This prospective study involved patients with T1D or T2D who underwent upper abdominal 3-T MRI. Additionally, patients without impaired glucose metabolism were retrospectively included as a control. The imaging data included pancreatic anteroposterior (AP) diameter, pancreas-to-muscle signal intensity ratio (SIR) on fat-suppressed T1-weighted image (FS-T1WI), apparent diffusion coefficient (ADC) value, T1 value on T1 map, proton density fat fraction (PDFF), and mean secretion grade of pancreatic juice flow on cine-dynamic magnetic resonance cholangiopancreatography (MRCP). The MR measurements were compared using one-way analysis of variance and the Kruskal–Wallis test. Results: Sixty-one patients with T1D (*n* = 7) or T2D (*n* = 54) and 21 control patients were evaluated. The pancreatic AP diameters were significantly smaller in patients with T1D than in patients with T2D (*p* < 0.05). The average SIR on FS-T1WI was significantly lower in patients with T1D than in controls (*p* < 0.001). The average ADC and T1 values of the pancreas were significantly higher in patients with T1D than in patients with T2D (*p* < 0.01) and controls (*p* < 0.05). The mean secretion grade of pancreatic juice flow was significantly lower in patients with T1D than in controls (*p* = 0.019). The average PDFF of the pancreas was significantly higher in patients with T2D than in controls (*p* = 0.029). Conclusions: Patients with T1D had reduced pancreas size, increased pancreatic T1 and ADC values, and decreased pancreatic juice flow on cine-dynamic MRCP, whereas patients with T2D had increased pancreatic fat content.

## 1. Introduction

Diabetes is a chronic metabolic disorder characterized by hyperglycemia resulting from insulin deficiency, insulin resistance, or a combination of both. It has many subclassifications, but the vast majority of diabetes cases can be divided into two main categories: type 1 diabetes (T1D) and type 2 diabetes (T2D), each with a different pathophysiology, presentation, and management [1].

T1D, which constitutes only 5–10% of diabetes, is characterized by an absolute insulin deficiency resulting from cellular-mediated autoimmune destruction of β-cells in the pancreas. In contrast, T2D, which accounts for 90–95% of those with diabetes, involves a complex interplay between genetics and environmental factors and ranges from predominantly insulin resistance with relative insulin deficiency to predominantly impaired insulin secretion with insulin resistance [1,2]. In both categories, pancreatic exocrine insufficiency, which is defined by a deficiency of pancreatic enzymes resulting in maldigestion [3], can also be observed (26–74% of patients with T1D and 10–56% of patients with T2D), and its mechanism is associated with multiple factors, including some histopathological changes, such as pancreatic atrophy and fibrosis [4].

In recent years, there have been significant advancements in image-based diagnostics and treatments in the oncological field [5,6]. Beyond conventional imaging techniques, such as computed tomography (CT) and magnetic resonance imaging (MRI), a new image analysis technique based on algorithms that allow for the extraction of imaging features from radiological images, known as radiomics, has been applied to improve diagnosis, prognostication, and treatment planning with the aim of delivering personalized medicine [7,8,9]. In the field of diabetes, a few studies using CT have found that the radiomics signature of the pancreas shows the highest discriminatory ability for early diabetes screening and prediction of T2D [10,11]. However, CT is not suitable for repeated measurements in young populations, such as patients with T1D, due to the high radiation dose associated with abdominal imaging. Therefore, it is preferable to utilize other noninvasive modalities, such as MRI, for image analysis in diabetes.

The utility of MRI to detect changes in images reflecting pancreatic histological and pathophysiological alterations in patients with diabetes has been increasingly explored [12,13]. For example, previous studies have shown that pancreatic volume measurements using MRI are reduced in patients with both T1D and T2D [14]. In studies of individuals with T2D, the T1 value of the pancreas on the T1 map < images, suggested as a marker of fibrosis, was found to be positively correlated with glycosylated hemoglobin (HbA1c) [15]. However, diabetes-related pancreatic changes on MRI, especially those in patients with T1D and differences from those in patients with T2D, remain unclear. In addition, most previous studies evaluated pancreatic images with only one or two limited MRI parameters or included limited clinical data.

Therefore, the present study evaluated the pancreatic changes on MRI in both T1D and T2D patient groups and clarified the differences in the imaging findings of the pancreas between these categories of patients versus those without diabetes as a control group < using multiparametric MRI.

## 2. Materials and Methods

### 2.1. Study Population

The institutional review board of our hospital approved this prospective study, and written informed consent was obtained from all patients before the investigation. We recruited consecutive eligible patients with T1D or T2D who were able to undergo upper abdominal MRI at the outpatient clinic of our hospital between January 2021 and January 2023. Glycemic status had been confirmed to meet the criteria for diabetes defined by the Japan Diabetes Society [16]. For T1D, the patients had reached an insulin-dependent state with autoantibodies against islet antigens that had been confirmed to arise since the onset or during the course of the disease. The inclusion criteria for patients with T1D and patients with T2D were as follows: (i) age of 40–75 years old; (ii) HbA1c ≤ 11%; and (iii) a body mass index (BMI) of 16–37 kg/m^2^. Patients were excluded from this study if any of the following conditions were diagnosed at registration: (i) renal failure (estimated glomerular filtration rate < 30 mL/min/1.73 m^2^); (ii) serious liver disease (liver cirrhosis or aspartate aminotransferase [AST] and aspartate transaminase [ALT] ≥ 100 IU/L); (iii) serious heart disease (i.e., heart failure or unstable angina pectoris); and (iv) contraindications for MRI. In addition, patients who were administered anti-cancer drugs and glucocorticoids and those with a medical history of hepatobiliary-gallbladder disease, gastrointestinal surgery, exocrine insufficiency or pancreatitis, or excessive alcohol consumption were excluded from this study.

For T1D, patients with established T1D were enrolled in this study. Patients within one year of T1D onset (recent onset) and those who had been diagnosed with fulminant T1D were excluded. Patients with T2D who had been under medical management for at least three years were also enrolled in order to include patients with established T2D and exclude patients with pancreatic cancer-associated diabetes. Patients diagnosed with diabetes before 25 years old and those who had a family history of diabetes in the past 3 generations were not enrolled in order to preclude maturity-onset diabetes of the young (MODY). Other etiologies of diabetes were excluded based on the following: (i) the absence of acanthosis nigricans or polycystic ovary syndrome and (ii) the absence of features consistent with maternally inherited diabetes and deafness or a renal cyst and diabetes syndrome (MODY5).

As a control group, we retrospectively included 21 patients without an impaired glucose metabolism (confirmed by HbA1c value), a history of diabetes, and pancreatic diseases, excluding pancreatic cysts, who met the above inclusion criteria for the age and BMI and underwent upper abdominal MRI during the same period of inclusion.

### 2.2. Clinical Data

For patients with diabetes, the following clinical information and laboratory test results were recorded at the time of MRI examination. Clinical information included the age, gender, age at diabetes onset, duration of diabetes, and BMI. Laboratory tests were conducted under fasting conditions on the same day of imaging as follows: HbA1c, fasting plasma glucose (FPG), C-peptide immunoreactivity (CPR), and glucagon for pancreatic endocrine function tests; P-amylase and lipase for pancreatic exocrine function tests; triglyceride (TG), high-density lipoprotein-cholesterol (HDL-C), and low-density lipoprotein-cholesterol (LDL-C) for lipid tests; AST and ALT for liver function tests; creatinine (Cr) and urine albumin for renal function tests. Plasma glucagon levels were measured using a sandwich ELISA (Mercodia AB, Uppsala, Sweden). The C-peptide index (CPI) was calculated using the following formula: fasting CPR/FPG [17]. For control patients, clinical data included the age, gender, BMI, and HbA1c at the time of MRI examination. These clinical data were extracted from the electronic clinical records.

### 2.3. MRI Technique

MRI was performed using a 3-T clinical MRI system (Vantage Centurian, Canon Medical Systems, Tochigi, Japan) equipped with a 16-channel body coil. The patients fasted for at least 4 h before MRI. Our routine protocol for pancreatic MRI included in-phase (IP) and opposed-phase (OP) two-dimensional (2D) T1-weighted gradient-echo (GRE) images, fat-suppressed three-dimensional (3D) T1-weighted GRE images, apparent diffusion coefficient (ADC) maps obtained by axial free-breathing diffusion-weighted imaging with a single-shot echo-planar sequence, T1 maps obtained using the modified Look–Locker method, proton density fat fraction (PDFF) maps obtained by axial breath-hold six-point Dixon imaging, and cine-dynamic MR cholangiopancreatography (MRCP) with a spatially selective inversion recovery (IR) pulse. The imaging parameters of each MRI sequence are listed in Table 1. IP and OP images were obtained to measure the pancreatic diameter. Fat-suppressed T1-weighted images (FS-T1WI), ADC maps, and T1 maps were obtained for assessing the pancreatic fibrosis [18,19,20]. PDFF maps were obtained to evaluate the pancreatic fat content. For the assessment of pancreatic exocrine function, cine-dynamic MRCP with a spatially selective IR pulse was performed [21].

For cine-dynamic MRCP with a spatially selective IR pulse, a breath-hold, thick-slab MRCP image was obtained as a reference using a fast advanced spin-echo sequence in the coronal plane. Next, using the same MR sequence, a spatially selective IR pulse (inversion time, 2200 ms) with a width of 20 mm was placed as perpendicular as possible to the main pancreatic duct to nullify the static pancreatic juice signal. Within the area of the spatially selective IR pulse, the static pancreatic juice showed a low signal intensity, whereas the secretory flow of the pancreatic juice showed a high signal intensity (Figure 1). MRCP with a spatially selective IR pulse was repeatedly performed every 15 s (with 5 s of scanning and 10 s of rest) for 5 min (20 images in total), and a series of these MRCP images was referred to as cine-dynamic MRCP with a spatially selective IR pulse [22].

### 2.4. Image Analyses

Two radiologists (M.H. and M.T., with 9 and 22 years of clinical experience, respectively) evaluated the images in consensus. The radiologists were blinded to the patients’ clinical information. Measurements of the pancreatic anteroposterior (AP) diameter were performed at the pancreatic head, body, and tail on OP images [23]. The radiologists measured the ADC value and PDFF of the pancreatic head, body, and tail on the ADC and PDFF map < images using operator-defined circular regions of interest (ROIs). The ROI circles were drawn as large as possible in a homogeneous area of the pancreas while avoiding the main pancreatic duct, focal lesions, and artifacts. T1 values were also measured at the pancreatic head and two locations on the pancreatic body or tail on the T1 map < images. In addition, they measured the signal intensities (SIs) of the pancreatic head, body, tail (SI_pancreas_), and paraspinal muscle (SI_muscle_) using FS-T1WI to calculate the SI ratio (SIR: SI_pancreas_/SI_muscle_). For these measurements, average values were used for the data analysis.

Cine-dynamic MRCP images were assessed independently by 2 radiologists (M.H. and K.I., with 9 and 36 years of clinical experience, respectively). Any discrepancies between reviewers were resolved by consensus. The MR images were assessed for (a) the frequency of the pancreatic juice flow observed in the main pancreatic duct over the 5-min imaging period (20 images) and (b) the secretion grade of the pancreatic juice, which was categorized based on the movement distance of the pancreatic juice flow within the area of a spatially selective IR pulse using a 5-point secretion grade score (grade 0, no flow; grade 1, <5 mm; grade 2, 5–10 mm; grade 3, 11–15 mm; grade 4, >15 mm) [22]. The mean secretion grade of the pancreatic juice was calculated as follows: (total grade score)/20.

### 2.5. Statistical Analyses

Statistical analyses were performed by using the SPSS software program (version 27.0; IBM, Armonk, NY, USA). Normality was assessed using the Shapiro–Wilk test. The differences in clinical data between the T1D and T2D groups were evaluated by two-sample *t*-test for FPG, LDL-C, and Cr (normally distributed data) and Mann–Whitney *U* test for age at diabetes onset, duration of diabetes, CPR, CPI, glucagon, P-amylase, lipase, TG, HDL-C, AST, ALT, and urine albumin (non-normally distributed data). Fisher’s exact test was used to assess the differences in gender among the T1D, T2D, and control groups. The differences in clinical data and MR measurements among the T1D, T2D, and control groups were assessed using one-way analysis of variance with Tukey’s post hoc test for BMI and pancreatic AP diameter (normally distributed data), and Kruskal–Wallis test with Dunn’s post hoc test for age, HbA1c, SIR on FS-T1WI, ADC value, T1 value, PDFF and cine-dynamic MRCP findings (non-normally distributed data). The correlations between the MRI measurements in patients with diabetes were evaluated using Spearman’s rank correlation coefficient analysis. Statistical significance was set at *p* < 0.05. To determine the magnitude of differences in MR measurements among the T1D, T2D, and control groups, effect sizes were calculated using partial eta-squared (η^2^) for the one-way analysis of variance and r for the non-parametric test. The effect sizes were interpreted as small (0.01–0.05), medium (0.06–0.13), and large (≥0.14) for η^2^, and as small (0.10–0.29), medium (0.30–0.49), and large (≥0.50) for r. The interobserver agreement between the 2 reviewers was assessed using weighted Cohen’s κ statistics and interpreted as follows: 0.21–0.40, fair agreement; 0.41–0.60, moderate agreement; 0.61–0.80, substantial agreement; and 0.81–1.00, excellent agreement.

## 3. Results

### 3.1. Patient Characteristics

A total of 67 patients with diabetes met the inclusion criteria. Among these patients, six were excluded because of suboptimal image quality on ADC maps, PDFF maps, or T1 maps. Ultimately, 61 patients with T1D (n = 7) or T2D (n = 54) and 21 control patients were enrolled in this study. Missing data were identified randomly in several clinical variables as follows: urine albumin (n = 1) in T1D; CPR (n = 1), CPI (n = 1), glucagon (n = 1), P-amylase (n = 1), lipase (n = 1) and urine albumin (n = 6) in patients with T2D. In the comparison of each clinical variable between the groups, cases with missing data were excluded from the analysis. The patient characteristics are summarized in Table 2.

The patient age was significantly lower in the T1D group < than in the T2D (*p* = 0.016) and control (*p* = 0.004) groups, while no significant differences were found in the patient age between the T2D and control groups (*p* = 0.799). The BMI was significantly higher in the T2D group < than in the T1D (*p* = 0.033) and control groups (*p* < 0.001), while there were no significant differences in the BMI between the T1D and control groups (*p* = 1.000). In the comparison between the T1D and T2D groups, the T1D group < had lower levels of CPR, CPI, P-amylase, lipase, TG, AST, ALT, and urine albumin than the T2D group (*p* < 0.05) but higher HDL-C levels (*p* = 0.001). No significant differences were observed in the gender, age at diabetes onset, duration of diabetes, HbA1c, FPG, glucagon, LDL-C, or Cr between the two diabetes groups.

### 3.2. The Comparison of MR Measurements Among the T1D, T2D, and Control Groups

A comparison of MR measurements among the three groups is shown in Figure 2 and Figure 3, and Table 3. The pancreatic AP diameters in the T1D group < were significantly smaller than those in the T2D group (*p* < 0.01). The AP diameters of the pancreatic body and tail in the T1D group < were also significantly smaller than those in the control group (*p* < 0.05). The η^2^ values for the pancreatic AP diameters ranged from 0.13 to 0.16, indicating a medium to large effect size. The average SIR on FS-T1WI in the T1D group < was significantly lower than that in the control group (*p* < 0.001, large effect size [*r* = 0.67]), and the SIR of the pancreatic body on FS-T1WI in the T1D group < was significantly lower than that in the T2D group (*p* = 0.008, medium effect size [*r* = 0.39]). The average ADC and T1 values of the pancreas in the T1D group < were significantly higher than those in the T2D (*p* < 0.001, large effect size [*r* = 0.50], and *p* = 0.001, medium effect size [*r* = 0.43], respectively) and control (*p* = 0.015, large effect size [*r* = 0.59], and *p* = 0.010, large effect size [*r* = 0.65], respectively) groups. No significant differences were observed in the PDFF of the pancreas between the T1D group < and both the T2D and control groups. In cine-dynamic MRCP, the interobserver agreement for the frequency of the detection of pancreatic juice flow and the grading of the moving distance of the pancreatic juice flow was excellent (κ value = 0.942, 95% confidence interval of 0.929–0.955). The frequency and mean secretion grade of pancreatic juice flow in the T1D group < were significantly lower than those in the control group (*p* = 0.016, medium effect size [*r* = 0.47] and *p* = 0.019, medium effect size [*r* = 0.49], respectively).

Regarding the MRI measurements in the T2D group, the average SIR on FS-T1WI was significantly lower (*p* = 0.006, medium effect size [*r* = 0.36]) and the average PDFF of the pancreas significantly higher (*p* = 0.029, medium effect size [*r* = 0.31]) than in the control group. However, no significant differences were found in the pancreatic AP diameters, ADC and T1 values of the pancreas, or the frequency and mean secretion grade of the pancreatic juice flow between the T2D and control groups.

### 3.3. The Correlation Between MR Measurements in Patients with T1D and T2D

The correlation between the MR measurements in patients with diabetes is shown in Table 4. The AP diameters of the pancreatic body and tail showed significant negative correlations with the T1 values of the pancreas (*r* = −0.407, *p* = 0.001 and *r* = −0.342, *p* = 0.007, respectively). The AP diameter of the pancreatic head showed a significant positive correlation with the PDFF of the pancreas (*r* = 0.330, *p* = 0.009). The SIR of the pancreatic head and body on FS-T1WI showed significant negative correlations with the T1 values (*r =* −0.478, *p* < 0.001 and *r* = −0.291, *p* = 0.023, respectively) and PDFF of the pancreas (*r* = −0.360, *p* = 0.004 and *r* = −0.262, *p* = 0.042, respectively), and also showed significant positive correlations with the mean secretion grade of pancreatic juice flow (*r* = 0.293, *p* = 0.022 and *r* = 0.392, *p* = 0.002, respectively). The ADC values of the pancreatic body and tail showed significant negative correlations with the PDFF of the pancreas (*r* = −0.274, *p* = 0.032 and *r* = −0.263, *p* = 0.043, respectively). The T1 value of the pancreatic head showed a significant positive correlation with the PDFF of the pancreas (*r* = 0.276, *p* = 0.031). There were no significant correlations among the other variables.

## 4. Discussion

Regarding the patient characteristics in this study, patients with T1D were significantly younger than T2D and control patients, which may be primarily attributed to the early onset and diagnosis of T1D [1]. The results of pancreatic function tests showed that the levels of CPR, CPI, P-amylase, and lipase were lower in patients with T1D than in patients with T2D, but the levels of HbA1c and FPG were not different between patients with T1D and patients with T2D. These observations indicate that glycemic status was not different in both patients, but T1D had lower levels of insulin and pancreatic digestive enzymes. Our results related to lipids showed that the TG level was higher and the HDL-C level was lower in patients with T2D than in patients with T1D. These findings suggest that patients with T2D were more associated with dyslipidemia, and they also had a high BMI compared to T1D and control patients. As for the liver and renal functions, the levels of AST, ALT, and urine albumin were higher in patients with T2D than in patients with T1D, suggesting that the liver and renal functions were more likely to be impaired in patients with T2D than in patients with T1D, although there was no marked decline in these functions in patients with T2D.

Our findings showed that the pancreas size was significantly reduced in patients with T1D compared to patients with T2D and control patients. Pancreatic size is known to decrease with age [24], but we found that patients with T1D in this study had significantly smaller pancreas diameters despite being significantly younger than T2D and control patients. In contrast, no significant difference was observed in pancreas size between the T2D and control groups. Most studies have demonstrated that the pancreas size is decreased in both patients with T1D and patients with T2D and that this change in the pancreas is more prominent in patients with T1D than in patients with T2D [13,14]. The reduction in pancreas size is associated with the lack of insulin-trophic effects on acinar cells due to insulin deficiency [25] as well as being a consequence of immunological destruction of the exocrine pancreas [26]. Therefore, our study findings may be explained by the differences in pathophysiology between T1D and T2D, supported by the clinical data of decreased C-peptide values and pancreatic digestive enzymes in patients with T1D and the relative preservation of both in patients with T2D.

Furthermore, the frequency and mean secretion grade of pancreatic juice flow were significantly decreased in patients with T1D, while those in patients with T2D were not significantly different from those in control patients. A previous study has shown that cine-dynamic MRCP with a spatially selective IR pulse has the potential to estimate pancreatic exocrine function and that a cutoff value of 0.7 for the mean secretion grade is the criterion for pancreatic exocrine insufficiency [21]. Thus, our findings suggest that patients with T1D had an impaired pancreatic exocrine function probably due to structural and functional reduction of acinar and beta-cell, whereas patients with T2D with their relatively well-preserved secretion of pancreatic digestive enzymes and insulin in this study had a preserved pancreatic exocrine function.

In this study, the SIR on FS-T1WI was significantly reduced in patients with T1D and T2D compared to control patients. In addition, the T1 and ADC values of the pancreas were significantly increased in patients with T1D compared to those in T2D and control patients, whereas no marked differences were found in the T1 or ADC values of the pancreas between T2D and control patients. Previous studies have demonstrated that pancreas-to-muscle SIR on T1-weighted imaging and ADC values of the pancreas are associated with pancreatic fibrosis [18,19]. T1 values on T1 mapping have also been correlated with histological measurements of fibrosis [20] and have been reported to be associated with chronic inflammation in the pancreas [27]. In addition, the ADC values reflect an amalgamation of cell density, cell membrane integrity, and viscosity in biological tissue [12]. Based on these data, our findings suggest a significant decrease in pancreatic cellular density and markedly increased pancreatic fibrosis or chronic inflammation in patients with T1D and mild changes in patients with T2D. Some results of our study are not compatible with those of previous studies [15,28,29], showing that the T1 and ADC values of the pancreas were significantly increased in patients with T2D. The pathophysiology of T2D is heterogeneous and depends on the disease stage [1]. In addition, some previous studies included patients with T2D according to HbA1c values, and the clinical backgrounds of these patients have not been well documented. Thus, the differences in the clinical background characteristics of patients with T2D may explain the discrepancies between our results and those of previous studies.

Regarding pancreatic fat content, the PDFF of the pancreas was significantly increased in patients with T2D, which might reflect a paracrine effect of insulin that can induce fat deposition in the pancreatic parenchyma. This finding is consistent with previous studies using MRI, which found a higher pancreatic fat content in patients with T2D than in control patients [14,30,31,32,33]. Furthermore, a fatty pancreas can be associated with obesity and dyslipidemia [33,34,35]. Therefore, the relatively high BMI and TG values and relatively low HDL-C values in patients with T2D in this study may also explain our results. Conversely, the PDFF of the pancreas in patients with T1D was not significantly different from that in control patients. Our findings suggest that pancreatic fat content may not be altered in patients with T1D, which is in agreement with the results of a previous study on children with T1D [36]. However, pancreatic fat content has been reported to increase with aging [37]. Thus, the significantly lower age of patients with T1D compared to control patients may have contributed to the absence of increased pancreatic fat content in patients with T1D.

In patients with diabetes, we found that the size of the pancreatic body and tail tended to decrease with the increase in the T1 values associated with pancreatic fibrosis. This result may be supported by the fact that diabetes has a harmful impact on the progression of chronic pancreatitis by aggravating fibrosis, leading to pancreatic atrophy [38], and suggests that fibrosis-induced pancreatic atrophy in diabetes, as seen in patients with T1D in this study, may occur more prominently in the body and tail of the pancreas. On the other hand, the size of the pancreatic head tended to increase with increased PDFF of the pancreas. Previous studies have shown that fatty infiltration is usually most prominent in the pancreatic head [39], and the AP diameter of the pancreatic head in patients with intense fatty infiltration is larger than that in the control group [40]. Our finding suggests that this change in the pancreatic head can occur in patients with diabetes as well. In addition, our findings showed that the PDFF of the pancreas had a significant negative correlation with the SIR on FS-T1WI and a significant positive correlation with the T1 value, indicating that the T1 signal intensity of the pancreas was likely to decrease with increased pancreatic fat content. A normal pancreas has an intrinsically high T1 signal intensity due to the T1-shortening effect of acinar protein within the pancreatic parenchyma [41]. Thus, our findings may be explained by the loss of acinar parenchyma associated with fatty infiltration in the pancreatic parenchyma in diabetes. Furthermore, the ADC values of the pancreas, the other suggested marker of fibrosis, showed a tendency to decrease with increased PDFF. This finding suggests that pancreatic fibrosis may be less likely to increase in cases of increased pancreatic fat content, as seen in patients with T2D in this study.

Regarding pancreatic exocrine function, we observed significant positive correlations between the mean secretion grade of pancreatic juice flow and the SIR on FS-T1WI associated with pancreatic fibrosis. Our findings suggest that pancreatic exocrine function may decrease with the progression of pancreatic fibrosis, which may be supported by previous studies demonstrating the increased pancreatic fibrosis and pancreatic exocrine dysfunction in both T1D and T2D [42]. Although we observed no significant correlations between the mean secretion grade of pancreatic juice flow and PDFF of the pancreas, a significant degree of fatty infiltration in the pancreas can be associated with severe exocrine pancreatic insufficiency [43]. Thus, further studies including diabetic patients with marked pancreatic fat deposition will be needed to evaluate the association between pancreatic fat content and pancreatic exocrine function in patients with diabetes.

In current clinical practice, laboratory tests are the cornerstone of the diagnosis and follow-up < of diabetes, and MRI has still had a limited impact on the management of diabetes. However, blood tests are subject to some important limitations and cannot fully reflect the pathophysiological features of diabetes [44]. Recent studies have evaluated the longitudinal changes in pancreas volume and shape in patients with T1D using MRI [45,46]. Virostko et al. [46] have shown that pancreatic volume measurements have the potential to predict the progression of T1D. Another study has demonstrated that the risk of onset of T2D increases with smaller pancreatic diameter and higher angle of pancreaticobiliary junction [47]. These studies suggest the potential utility of MRI for monitoring the progression of diabetes and predicting its risk factors. In the present study, we demonstrated the MR imaging features of the pancreas reflecting the pancreatic endocrine and exocrine functions as well as pancreatic pathohistological changes in both patients with T1D and patients with T2D. Thus, multiparametric MRI may have the potential to improve our understanding of pancreatic changes in diabetes patients. Moreover, the application of multiparametric MRI to advanced image analysis techniques, such as radiomics, might have a potential role in improving diagnosis, monitoring disease progression, and assessing treatment response in these patients noninvasively. However, our study findings need to be validated by further clinical investigation using a large number of patients with diabetes. Additionally, it is necessary to assess the MR imaging findings over a long period and evaluate the correlation between MR imaging features, clinical findings, and treatments for future clinical applications.

Several limitations associated with the present study warrant mention. First, the number of patients with T1D was small due to the limited number of patients at our institution and our criteria for excluding genetic types of diabetes, which may have caused selection bias and may impact the generalizability and robustness of the imaging findings in patients with T1D. Moreover, the small sample size of patients with T1D and unbalanced sample size between groups may lead to reduced power, lower reproducibility, and lower accuracy in this study. The small effect size in some analyses may have been impacted by these factors. Second, the age of patients with T1D was significantly lower than that of T2D and control patients, and previous studies have shown that aging can affect the pancreas size, pancreatic fat content, pancreatic fibrosis, and pancreatic exocrine function [37,48,49]. Thus, further studies with a larger population with T1D and the T2D and control patients adjusted for age and sample size will be required to validate our results by recruiting further diabetes patients and volunteer subjects in our institution and conducting a multicenter study. Third, we did not measure fecal elastase-1 (FE-1) in spot stool, which is the gold standard test for the assessment of pancreatic exocrine function. The evaluation of pancreatic exocrine function using cine-dynamic MRCP with a spatially selective IR pulse is an indirect method. Furthermore, the BT-PABA test, used as a reference standard for comparison with cine-dynamic MRCP, has inferior diagnostic ability for pancreatic exocrine function compared to the FE-1 test [50]. Therefore, further validation using the FE-1 test as the reference standard is needed to establish cine-dynamic MRCP with a spatially selective IR pulse as a useful method for assessing the pancreatic exocrine function. Additionally, we measured the PDFF, T1, and ADC values using only a single slice for each part of the pancreas by ROI. This ROI-based measurement may induce measurement bias in a mosaic tissue such as the pancreas. For this reason, it is necessary to evaluate these images independently by multiple reviewers and measure these values using multiple slices in the future study. Finally, the imaging findings of the pancreas in patients with T2D may vary depending on the stage and duration of diabetes or treatment. Therefore, further clinical studies are needed to consider the clinical background of patients with T2D.

## 5. Conclusions

Patients with T1D had a reduced pancreas size, increased pancreatic T1 and ADC values suggestive of increased pancreatic fibrosis and decreased pancreatic cellular density, and decreased pancreatic exocrine function estimated by cine-dynamic MRCP with a spatially selective IR pulse, whereas patients with T2D had an increased pancreatic fat content. Multiparametric MRI of the pancreas provided characteristic imaging findings in patients with T1D and patients with T2D and was considered clinically helpful in understanding the pathophysiology of these diseases.

## Figures and Tables

**Figure 1 tomography-11-00016-f001:**
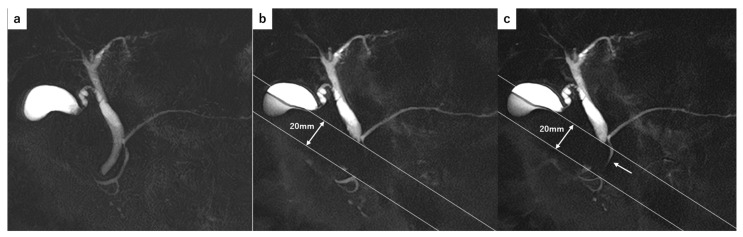
(**a**) MRCP image without a spatially selective IR pulse obtained as a reference image. (**b**) The static pancreatic juice within the area of the spatially selective IR pulse (the area of 20 mm width between the parallel white lines) showed a low signal intensity. (**c**) The pancreatic juice flow showed a high signal intensity (arrow) within the area of the IR pulse. The grade score of the pancreatic juice flow was classified as grade 3 (11–15 mm).

**Figure 2 tomography-11-00016-f002:**
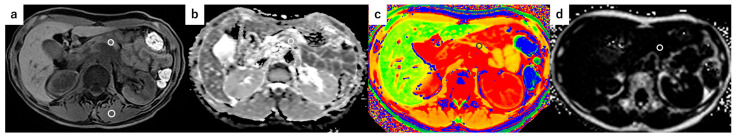
MR images from a 40-year-old woman with type 1 diabetes. On the fat-suppressed T1-weighted image (**a**), the average pancreas-to-muscle SIR was 1.08. On the ADC map (**b**), averaged ADC value of the pancreas was 1.86 × 10^−3^ mm^2^/s. On the T1 map (**c**), the averaged T1 value of the pancreas was 952 ms. On the PDFF map (**d**), the average PDFF of the pancreas was 2.2%.

**Figure 3 tomography-11-00016-f003:**
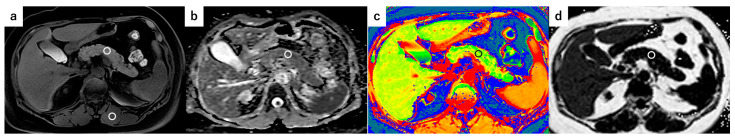
MR images from a 53-year-old woman with type 2 diabetes. On the fat-suppressed T1-weighted image (**a**), the average pancreas-to-muscle SIR was 1.45. On the ADC map (**b**), averaged ADC value of the pancreas was 1.03 × 10^−3^ mm^2^/s. On the T1 map (**c**), the average T1 value of the pancreas was 719 ms. On the PDFF map (**d**), the average PDFF of the pancreas was 8.6%.

**Table 1 tomography-11-00016-t001:** MRI sequence parameters.

	IP/OP	FS-T1WI	ADC Map	T1 Map	PDFF Map	Cine-Dynamic MRCP
TR (ms)	125	3.4	1663	5.3	7.8	5000
TE (ms)	2.52/1.50	1.5	51	1.8	1.2, 2.2, 3.2, 4.2, 5.2, 6.2	507
acquisition matrix	224 × 288	256 × 288	200 × 128	192 × 256	144 × 160	512 × 512
FOV (mm)	300 × 360	320 × 380	280 × 380	320 × 360	300 × 360	350 × 350
flip < angle (°)	70	12	90	13	3	150
slice thickness (mm)	5.0	1.5	5.0	8	2.3	50
band width (Hz/pixel)	976.5	781	3125	488	1302	558
interslice gap (mm)	1.4	-	0.8	-	-	-
parallel imaging factor	1.5	2.5	2	2	2	2
acquisition time (s)	40 (20 × 2)	20	85	34 (17 × 2 slice)	21	-
b-factor (s/mm^2^)	-	-	0, 1000	-	-	-

TR, repetition time; TE, echo time; FOV, field of view; IP, in-phase; OP, opposed-phase; FS-T1WI, fat-suppressed T1-weighted image; ADC, apparent diffusion coefficient; PDFF, proton density fat fraction; MRCP, magnetic resonance cholangiopancreatography.

**Table 2 tomography-11-00016-t002:** Patient characteristics.

	T1D		T2D		Controls		*p* Value
	n	Value	n	Value	n	Value	
Age (y) *	7	48 (43–51) ^a^	54	63 (54–70)	21	68 (59–70)	**0.005**
Gender: male (%) ^†^	2	29	29	54	10	48	0.540
Age at diabetes onset (y) ^‡^	7	32 (25.5–48.5)	54	47.5 (42–54)			0.072
Duration of diabetes (y) ^‡^	7	14 (6.5–21)	54	13.5 (8.3–19.5)			1.000
BMI (kg/m^2^) ^§^	7	22.0 ± 2.2	54	25.9 ± 3.9 ^b^	21	22.0 ± 3.7	**<0.001**
HbA1c (%) *	7	7.3 (6.65–7.8)	54	7.1 (6.6–8)	21	5.6 (5.5–5.9) ^c^	**<0.001**
FPG (mg/dL) ^||^	7	146 ± 43.8	54	135 ± 30.8			0.413
CPR (ng/mL) ^‡^	7	0 (0–0.85)	53	2 (1.52–2.82)			**<0.001**
CPI ^‡^	7	0 (0–0.63)	53	1.54 (1.09–2.11)			**<0.001**
Glucagon (pg/mL) ^‡^	7	14.3 (10.7–38.2)	53	24.7 (19.1–34.2)			0.276
P-Amylase (IU/l) ^‡^	7	12.5 (11.6–20.8)	53	29 (22–37.8)			**0.008**
Lipase (IU/l) ^‡^	7	18 (16–23)	53	37 (28–53)			**<0.001**
TG (mg/dL) ^‡^	7	70 (45–75)	54	126 (99–173)			**<0.001**
HDL-C (mg/dL) ^‡^	7	81 (65–88)	54	51 (45–64)			**0.001**
LDL-C (mg/dL) ^||^	7	114 ± 19	54	103 ± 26			0.329
AST (IU/L) ^‡^	7	15 (13–19)	54	22 (18–28)			**0.014**
ALT (IU/L) ^‡^	7	15 (11–17)	54	25 (17–31)			**0.007**
Cr (mg/dL) ^||^	7	0.69 ± 0.10	54	0.81 ± 0.23			0.192
Urine albumin (mg/g·Cr) ^‡^	6	8.1 (4.9–9.5)	48	15.0 (8.9–49.4)			**0.030**

Data are mean ± standard deviation or median with 25th and 75th percentiles in parentheses. T1D, type 1 diabetes; T2D, type 2 diabetes; BMI, body mass index; HbA1c, hemoglobin A1c; FPG, fasting plasma glucose; CPR, C-peptide immunoreactivity; CPI, C-peptide index; TG, triglyceride; HDL-C, high-density lipoprotein-cholesterol; LDL-C, low-density lipoprotein-cholesterol; AST, aspartate aminotransferase; ALT, aspartate transaminase; Cr, creatinine; * Kruskal–Wallis test, † Fisher’s exact test, ‡ Mann–Whitney *U* test, § One-way analysis of variance, || Two-sample *t*-test. ^a^ Value was significantly lower (*p <* 0.05) than that in T2D and controls. ^b^ Value was significantly higher (*p <* 0.05) than that in T1D and controls. ^c^ Value was significantly lower (*p <* 0.001) than that in T1D and T2D. The values in bold denote statistical significance.

**Table 3 tomography-11-00016-t003:** Comparison of MR measurements among the T1D, T2D, and control groups.

	T1D (n = 7)	T2D (n = 54)	Controls (n = 21)	*p* Value				Effect Size *η^2^ r*			
Region	1	2	3	All	1–2	1–3	2–3	All	1–2	1–3	2–3
AP diameter (mm) *											
Head	19.1 ± 2.8	27.0 ± 5.9	24.1 ± 4.9	**0.001**	**0.002**	0.099	0.106	0.16			
Body	10.2 ± 2.3	17.0 ± 5.2	17.3 ± 5.3	**0.004**	**0.004**	**0.006**	0.964	0.13			
Tail	12.8 ± 4.7	20.0 ± 5.5	18.6 ± 4.4	**0.003**	**0.002**	**0.029**	0.566	0.14			
FS-T1WI SIR ^†^											
Averaged	1.24 (1.18–1.34)	1.41 (1.32–1.49)	1.52 (1.47–1.61)	**<0.001**	0.059	**<0.001**	**0.006**		0.32	0.67	0.36
Head	1.26 (1.04–1.30)	1.37 (1.28–1.46)	1.49 (1.46–1.62)	**<0.001**	0.061	**<0.001**	**0.004**		0.32	0.68	0.38
Body	1.25 (1.20–1.37)	1.49 (1.42–1.57)	1.52 (1.47–1.69)	**0.002**	**0.008**	**0.001**	0.583		0.39	0.64	0.15
Tail	1.23 (1.21–1.37)	1.37 (1.27–1.49)	1.50 (1.43–1.67)	**0.003**	0.533	**0.011**	**0.015**		0.17	0.54	0.33
ADC value (×10^−3^ mm^2^/s) ^†^											
Averaged	1.40 (1.34–1.55)	1.15 (1.09–1.23)	1.24 (1.13–1.31)	**<0.001**	**<0.001**	**0.015**	0.352		0.50	0.59	0.19
Head	1.48 (1.37–1.66)	1.09 (1.03–1.22)	1.18 (1.09–1.30)	**<0.001**	**<0.001**	0.058	0.103		0.47	0.53	0.26
Body	1.40 (1.40–1.58)	1.17 (1.11–1.28)	1.23 (1.13–1.36)	**<0.001**	**<0.001**	**0.013**	0.678		0.49	0.54	0.14
Tail	1.43 (1.26–1.50)	1.18 (1.09–1.27)	1.19 (1.13–1.32)	**0.036**	**0.033**	0.199	1.000		0.31	0.40	0.11
T1 value (ms) ^†^											
Averaged	902 (880–936)	777 (731–819)	793 (742–838)	**0.002**	**0.001**	**0.010**	1.000		0.43	0.65	0.08
Head	940 (903–977)	765 (730–830)	783 (750–842)	**<0.001**	**<0.001**	**0.008**	1.000		0.47	0.64	0.12
Body or tail	900 (830–944)	781 (726–812)	777 (742–836)	**0.010**	**0.008**	**0.045**	1.000		0.38	0.49	0.07
PDFF (%) ^†^											
Averaged	2.2 (2.0–6.9)	6.3 (4.0–8.8)	4.0 (2.5–5.5)	**0.013**	0.196	1.000	**0.029**		0.22	0.09	0.31
Head	3.7 (2.6–9.3)	8.8 (4.7–13.0)	4.2 (3.1–8.1)	**0.035**	0.385	1.000	0.058		0.18	0.04	0.27
Body	1.0 (0.9–6.2)	4.9 (3.4–7.2)	3.5 (2.5–4.7)	**0.032**	0.238	1.000	0.078		0.20	0.15	0.27
Tail	3.1 (1.8–5.1)	4.8 (3.2–8.8)	3.2 (1.8–4.8)	**0.019**	0.463	1.000	**0.024**		0.18	0.04	0.31
cine-dynamic MRCP ^†^											
Frequency of pancreatic juice flow	3 (0–9)	12 (5–16.8)	15 (12–19)	**0.013**	0.251	**0.016**	0.127		0.24	0.47	0.24
Mean secretion grade of pancreatic juice	0.15 (0–0.625)	0.825 (0.25–1.425)	1.05 (0.65–2.0)	**0.021**	0.156	**0.019**	0.327		0.25	0.49	0.19

Data are mean ± standard deviation or median with 25th and 75th percentiles in parentheses. T1D, type 1 diabetes; T2D, type 2 diabetes; AP, anteroposterior; FS-T1WI, fat-suppressed T1-weighted image; SIR, signal intensity ratio; ADC, apparent diffusion coefficient; PDFF, proton density fat fraction; MRCP, magnetic resonance cholangiopancreatography. * One-way analysis of variance with Tukey’s post hoc test, † Kruskal–Wallis test with Dunn’s post hoc test. The values in bold denote statistical significance.

**Table 4 tomography-11-00016-t004:** Correlation between MR measurements in patients with diabetes.

Variables		1	2	3	4	5	6
1. AP diameter	Head	1	−0.019	−0.120	−0.175	0.330 **	0.027
	Body		0.085	−0.239	−0.407 **	0.092	0.037
	Tail		−0.146	−0.185	−0.342 **	−0.017	0.051
2. FS-T1WI SIR	Head		1	−0.145	−0.478 ***	−0.360 **	0.293 *
	Body			−0.060	−0.291 *	−0.262 *	0.392 **
	Tail			0.170	−0.139	−0.238	0.198
3. ADC value	Head			1	0.233	−0.247	−0.063
	Body				0.168	−0.274 *	−0.102
	Tail				0.088	−0.263 *	−0.028
4. T1 value	Head				1	0.276 *	−0.123
	Body/Tail					0.171	−0.094
5. PDFF	Head					1	0.059
	Body						−0.065
	Tail						−0.057
6. Mean secretion grade							1

N = 61. * *p* < 0.05. ** *p* < 0.01. *** *p* < 0.001. AP, anteroposterior; FS-T1WI, fat-suppressed T1-weighted image; SIR, signal intensity ratio; ADC, apparent diffusion coefficient; PDFF, proton density fat fraction.

## Data Availability

The raw data supporting the conclusions of this article will be made available by the authors upon request.

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
