# Peer review of "Multiparametric Magnetic Resonance Imaging Findings of the Pancreas: A Comparison in Patients with Type 1 and 2 Diabetes"

_tomography, 2025, doi:10.3390/tomography11020016_

Round 1
Reviewer 1 Report
Comments and Suggestions for Authors
This is an interesting study. I have several minor comments for the authors to address:
1. For 2.1 Study population, the diagnosis criteria for the T1D and T2D should be included. The exclusion criteria were not com complete, because several subjects were excluded based on the criteria that not were listed here.
2. For the cine-dynamic MRCP exam, please provide more detailed rationale for this exam and explain what metrics are compared between the groups.
Reviewer 2 Report
Comments and Suggestions for Authors
Thank you for your invitation. This seems to be an interesting research project. Before I formally agree to accept your paper, you need to revise your article and carefully answer my questions.
Section 1 Introduction
In the introduction section, the definition of "exocrine pancreatic insufficiency" can be simplified or moved to a later discussion.
Section 2.1 Study population
Regarding the acquisition of clinical data, I hope you can specifically show the calculation process and explain its specific meaning. Also, in the subsequent results, please refer to the relevant parameters for explanation.
Section 2.3 MRI technique
In this section, your description of the parameters is overly detailed. I suggest you present them in a chart form to simplify the text and visually display your parameters. Additionally, you have not provided a detailed description of the basis for parameter selection, such as why T1 values and ADC values are considered indicators of pancreatic fibrosis or inflammation.
Section 3.1 Patient characteristics
When describing that the average age of T1D patients is significantly lower than that of other groups, please consider and mention potential confounding factors.
3.2. The comparison of MR measurements among T1D, T2D, and control groups
Your explanation of Table 1 "Patient Characteristics" is not detailed enough. It is hoped that each comparison result can be elaborated in separate paragraphs to clarify the significance of the presented results.
Section 4 Discussion
You could delve deeper into the possible mechanisms behind the increased pancreatic fat content in T2D patients, such as its relationship with BMI or insulin action. For the reduced pancreatic size in T1D patients, more details could be provided by integrating the mechanism of immune destruction. Clearly define the potential application scenarios of multi-parameter MRI in actual diagnosis or treatment, such as early screening or disease monitoring.
Reviewer 3 Report
Comments and Suggestions for Authors
The title “Multiparametric magnetic resonance imaging findings of the pancreas: A comparison in patients with type 1 and 2 diabetes” is appropriate for the content covered.
The manuscript discusses the evaluation of pancreatic changes on MRI in both T1D and T2D patient groups. It clarifies the differences in pancreas imaging findings between these categories of patients versus those without diabetes as a control group using multiparametric MRI. However, the manuscript has some shortcomings in the area of image analysis and validation of the performance of the proposed method.
The topic is very interesting and current, but the manuscript setting needs to be improved. To make the document more validated, some sections need to be substantially expanded, particularly following the comments.
Please see the comments below.
1) Authors are strongly encouraged to consider the following state of the art:
a)The authors are strongly encouraged to expand the introductory section also taking into account the diagnostic and curative aspects that are currently being advanced in cancers such as pancreatic, gastric and gastrointestinal cancers and based on images and precision medicine [10.3390/cancers16193323]; furthermore, it is strongly recommended to discuss also other types of image analysis based on artificial intelligence, in particular radiomics, based on algorithms that allow to extract more information from images, as in [10.1007/978-3-031-51026-7_9] where the authors evaluate the performance of MRI radiomics analysis in distinguishing Low Grade versus High Grade bladder lesions and muscle-invasive bladder cancer versus non muscle-invasive bladder cancer. Expanding these points within the manuscript would make the work much more impactful and much more current, in the perspective of a more predictive and applicative application in the field of personalized medicine and patient stratification. This will allow readers and users of this manuscript to reflect on the valuable contribution of theranostics and radiomic analysis in improving the characterization and differentiation of bladder lesions, both in terms of differentiation of cancer lesions and discrimination, diagnosis and treatment between different grades of pathology.
b)In [10.1111/j.1464-5491.2007.02027.x ] MRI was used to monitor pancreatic atrophy in type 1 diabetes; in [10.1016/j.pan.2014.04.031], To compare pancreas volume (PV) measurement using MRI-based planimetry in patients with Type 2 diabetes mellitus (DM) to PV in patients without diabetes, a regression model was used. The authors should expand and update the state of the art considering these high impact works to give the manuscript a greater applicability also through the methods proposed by other works of the same caliber, to make more effective the discussions based on the state of the art currently available.
These suggestions will be essential and are recommended to the authors so that the limitations and strengths of the study are widely highlighted; furthermore this evaluation of the validity of the methods, results and interpretation of the data will bring a higher scientific impact of this promising work.
2) A list of abbreviations at the end of the manuscript would help make the manuscript more explanatory. Authors are requested to include them.
3) Authors are strongly recommended to include a graphical or tabular summary of the statistical analysis performed, clarifying which methods were adopted for reproducibility and accuracy and for performance evaluation of the proposed applied method.
4) English is quite understandable and does not require any particular improvement.
Finally, it would be helpful to extend the references to enhance the coherence of the article.
Reviewer 4 Report
Comments and Suggestions for Authors
Thank you for the opportunity to review this manuscript on multiparametric MRI findings comparing pancreatic changes in type 1 and type 2 diabetes. The authors present an ambitious investigation examining pancreatic alterations in type 1 and type 2 diabetes through multiparametric magnetic resonance imaging. While the research question holds significant clinical relevance and the technical approach demonstrates innovation, several fundamental limitations warrant careful consideration.
The manuscript's conceptual framework demonstrates sophistication in its approach to characterizing diabetes-associated pancreatic changes through advanced imaging modalities. The technical methodology, incorporating multiple MRI sequences including fat-suppressed T1-weighted imaging, diffusion-weighted imaging, T1 mapping, and dynamic MRCP, reflects commendable rigor in protocol design. However, critical methodological vulnerabilities compromise the study's validity. The profound numerical asymmetry between cohorts (T1D n=7, T2D n=54, Controls n=21) introduces insurmountable statistical constraints. While the inclusion of a control group provides valuable normative data, the minimal T1D sample size precludes robust comparative analyses. This limitation is compounded by significant demographic heterogeneity, particularly the age differential between groups, which introduces confounding variables that cannot be adequately controlled through standard statistical methods.
The imaging protocols demonstrate technical sophistication, yet the absence of established reference standards for pancreatic exocrine function (e.g., fecal elastase-1) limits validation of the MRCP findings. The observed patterns of pancreatic involvement - reduced dimensions and altered function in T1D versus predominantly increased fat content in T2D - while intriguing, must be interpreted with considerable caution given these methodological constraints.
The statistical approach requires substantial refinement. The authors must address:
- Inadequate power calculations
- Limited control for demographic confounders
- Absence of effect size estimations
- Insufficient handling of multiple comparisons
The discussion, while thoughtful in its engagement with existing literature, insufficiently addresses these fundamental limitations. The clinical implications and future directions require more robust development within the context of the study's constraints.
This investigation, despite its innovative approach and potential significance, falls short of the methodological rigor required for publication in its current form. Substantial revision addressing these critical limitations would be essential for meaningful contribution to the field of diabetes imaging.
The manuscript exemplifies the challenges inherent in studying differential disease manifestations with markedly uneven cohort sizes. These methodological constraints fundamentally limit the validity and generalizability of the findings, regardless of the sophisticated imaging approach employed.
Round 2
Reviewer 2 Report
Comments and Suggestions for Authors
Thank you for your invitation. This looks like an interesting piece of research. For this paper, I have the following overall suggestions:
1. Enhance the depth of data analysis and interpretation:
The paper presents a large amount of data in the "Results" section, including clinical data, MRI measurements, etc. However, the analysis and interpretation of these data are relatively brief in the "Discussion" section. It is suggested that the author further dig into the meaning behind the data, for example, to explore the relationship between different variables through more detailed statistical analysis (such as multiple regression analysis, correlation analysis, etc.).
In addition, significant differences between different types of diabetes and between control groups can be compared, and possible causes and biological mechanisms for these differences can be discussed. This will help readers better understand the link between MRI parameters and diabetes pathophysiology.
2. Increase the clinical relevance and practicability of the research:
The potential applications of MRI technology in diabetes management, such as as a non-invasive, objective means of monitoring, and possible directions for future research (such as combining artificial intelligence, machine learning and other technologies for more accurate diagnosis and prediction) can be explored. This will make the research more prospective and practical.
The paper gives a clear description of the research results.
Reviewer 3 Report
Comments and Suggestions for Authors
The authors have responded comprehensively to all comments and made the manuscript more complete and detailed in every way. The bibliography is also more complete and consistent now.
Author Response
Thank you so much for your feedback. We are glad to hear that our revised manuscript has been improved as per your suggestions.
Reviewer 4 Report
Comments and Suggestions for Authors
The authors' response to the critiques demonstrates a commendable effort to address fundamental methodological concerns, though certain aspects warrant further refinement. Their acknowledgment of the profound sample size disparity (T1D n=7, T2D n=54, Controls n=21) and subsequent incorporation of effect size analyses represents a significant methodological enhancement. The comprehensive tabulation of effect sizes using both η² and r metrics provides valuable context for interpreting their findings, particularly in light of the inherent statistical limitations.
However, the authors' treatment of demographic heterogeneity, specifically the age differential between cohorts, remains somewhat inadequate. While they forthrightly acknowledge their inability to implement age adjustments due to distributional disparities and insufficient T1D sample size, this limitation deserves more nuanced integration throughout their results interpretation, rather than mere acknowledgment in the limitations section.
The absence of fecal elastase-1 measurements as a reference standard for pancreatic exocrine function represents another methodological vulnerability. Though the authors provide a reasonable justification for employing cine-dynamic MRCP based on previous validation studies, this alternative approach would benefit from more robust validation data and explicit discussion of its limitations relative to established reference standards.
Particularly noteworthy is their inclusion of effect size calculations, which reveals medium to large effects for key imaging findings despite small effects in some analyses. This statistical enhancement provides crucial context for interpreting their results within the constraints of their sample size limitations. However, the manuscript would benefit from more thorough integration of these effect size considerations throughout the results section, rather than treating them as a separate methodological addendum.
The authors have made substantive progress in addressing the reviewer's concerns, though opportunities remain for deeper integration of these methodological considerations throughout the manuscript. Their candid acknowledgment of limitations, coupled with efforts to provide additional statistical context through effect size analyses, strengthens the scientific validity of their work despite the inherent constraints of their study design.
